# Integrating Ecosystem Services into Risk Assessments for Drinking Water Protection

Nadine Gärtner [1,*], Andreas Lindhe [1], Julia Wahtra [2], Tore Söderqvist [3], Lars-Ove Lång [1,4], Henrik Nordzell [5], Jenny Norrman [1] and Lars Rosén [1]

1 Department of Architecture and Civil Engineering, Chalmers University of Technology, 41296 Gothenburg, Sweden; andreas.lindhe@chalmers.se (A.L.); lars-ove.lang@sgu.se (L.-O.L.); jenny.norrman@chalmers.se (J.N.); lars.rosen@chalmers.se (L.R.)
2 Department of Economics, Swedish University of Agricultural Sciences, 75007 Uppsala, Sweden; julia.wahtra@slu.se
3 Holmboe & Skarp AB, 14896 Sorunda, Sweden; tore.soderqvist@holmboe-skarp.se
4 Geological Survey of Sweden, 41320 Gothenburg, Sweden
5 Ramboll Sweden AB, 11851 Stockholm, Sweden; henrik.nordzell@ramboll.se
* Correspondence: nadine.gartner@chalmers.se

**Abstract:** Water protection is a widely supported goal in society, but competing interests often complicate the implementation of water protection measures. Moreover, the benefits of protection efforts are typically underestimated as risk assessments focus on the provision of drinking water and neglect the additional services provided by a clean drinking water source. We developed a list of water system services (WSS) that allows assessment of all biotic and abiotic services provided by a drinking water source. The WSS were derived from the Common International Classification of Ecosystem Services (CICES). The objectives of this paper are to (i) introduce the concept of WSS, (ii) describe a procedure on how to develop a region-specific list of WSS and present a list of WSS specifically tailored to Sweden, (iii) present how to integrate WSS into a risk assessment for drinking water, and (iv) illustrate a practical application on a Swedish case study. The results, presented as an assessment matrix, show the provided services and contrast the hazard sources with their impact on all services. The WSS assessment can be used to communicate and negotiate the extent of water protection measures with relevant stakeholders and illustrate synergies and trade-offs of protective measures beyond drinking water protection.

**Keywords:** ecosystem services; water system services; drinking water; water protection; source water protection; risk assessment; water safety plan

## 1. Introduction

Reliable access to good quality water sources is key for human development. However, the world's drinking water sources are under growing pressure due to human activities, including infrastructure projects, agriculture, climate change, and the ever-increasing need for freshwater [1].

To promote reliable access to good quality water, the World Health Organization (WHO) recommends the development of risk assessments as part of Water Safety Plans [2]. These plans encompass a proactive risk assessment and management where all hazards towards the drinking water are mapped, potential risks estimated, and protective measures evaluated to support decision making. In recent years, the role and importance of a risk-based approach and the use of risk assessments as a basis for implementing water protection measures have been emphasized by various national authorities, e.g., in Sweden [3].

However, the application of conventional risk assessment methods for drinking water resources suffers from several challenges. Among them are the additional costs of conducting a risk assessment [4] and a scope of analysis that is limited to the service of drinking

water. With this limited scope, risk assessments may neglect other important services provided by drinking water sources. For example, an extension of a water protection area benefits the provision of drinking water while simultaneously promoting other services, e.g., using a lake for swimming, visiting a beautiful spring, watering livestock, or installing ground source heat pumps. Decisions based on conventional risk assessments neglect those important services which may justify additional protection measures.

One approach suitable to guide an expansion of the scope of risk assessments is the ecosystem services (ES) framework. ES are defined as the contributions ecosystems make to human well-being [5]. The ES framework provides an approach for managing natural resources and identifying nature's benefits to society.

However, applying the ES framework for decision making remains challenging despite decision-makers' interest in employing it in policy decisions [6,7]. Nevertheless, practitioners and decision-makers often lack the capacity and resources to incorporate an extensive ES assessment into their decision making [8]. Existing classification systems of ES comprise extensive catalogs listing all ES that ecosystems provide. While the comprehensiveness of such catalogs is crucial in a research context, it simultaneously hinders the use of the ES framework by practitioners. Carrying out an ES assessment using, e.g., the Common International Classification of Ecosystem Services (CICES) for each water source is likely to be impracticable, time-consuming, and financially not viable. Olander et al. [8] call for straightforward and cost-effective methods when including ecosystem service assessments in decision making. There is a need to make the ES framework more operational for practitioners and develop a more compiled version specifically for drinking water sources.

The overall aim of this study is to illustrate how the ES framework can be integrated into risk assessments for drinking water protection to ensure that the full range of services provided by the water source is accounted for in decision making. The objectives of this paper are as follows: first, we introduce the ES framework and the concept of water system services (WSS). Second, we explain the process of modifying the ES framework to develop a region-specific list of services based on the classification of ES contained in CICES [9]. Third, we explain the adaptation of risk assessments for drinking water to facilitate the use of ES. Lastly, we present the results of the practical application of the approach in a Swedish case study.

## 2. From Ecosystem Services to Water System Services

Throughout history, humans have always relied upon nature and well-functioning ecosystems. The contributions ecosystems make to human well-being can be defined as ecosystem services (ES). In the 1970s, Westman [10] linked human welfare to functioning ecosystems and formalized this relation. Two decades later, Daily [11], Costanza et al. [12], and the Millennium Ecosystem Assessment [13] contributed to mainstreaming the concept of ecosystem services. Since then, the number of ES studies has grown exponentially [14].

To assess ecosystem services thoroughly, various classification systems of ecosystem services have been developed (e.g., Common International Classification of Ecosystem Services (CICES) [9], the Economics of Ecosystems and Biodiversity (TEEB) [15], and the Millennium Ecosystem Assessment (MA) [13]). In particular, CICES is a standardized classification scheme broadly accepted, recognized, and applied in ecosystem services research in Europe [16,17]. Building on earlier classification schemes such as TEEB and MA, CICES uses a cascade model as a conceptual framework [18] in which a production chain links biophysical structures over various steps to the contribution to human well-being. In the natural system, biophysical structures and processes provide ecosystem functions that generate ecosystem services. The actual use of a service in the social and economic system then provides humans with benefits that can be valued.

The hierarchical structure applied in CICES resembles a taxonomic approach used for categorizing organisms. Services are divided into sections (provisioning, regulating, and cultural services), then into divisions, groups, classes, and finally providing examples of services belonging to different classes (see Table 1).



**Table 1.** Excerpt from CICES v5.1 illustrating the hierarchical structure used to describe services.

| Section | Division | Group | Class | Class Type | Code | Simple Descriptor | Ecological Clause | Use Clause | Example Service | Example Goods and Benefits |
|---|---|---|---|---|---|---|---|---|---|---|
| Provisioning (abiotic) | Water | Groundwater used for nutrition, materials, or energy | Ground and (sub-surface) water for drinking | By amount, type, source | 4.2.2.1 | Drinking water from the below ground | Natural, below ground-water bodies or aquifers … | … that provide a source of drinking water | Aquifer volume and char-acteristics | Potable water in public supply system; mineral water |

A growing number of studies have focused on the supply of ecosystem services by water bodies [17,19], but the demand for more specified uses remains large [7]. Within this body of literature, the link between ecosystem services and water bodies has previously been assessed in two ways. Studies either assessed water-related ecosystem services produced by terrestrial systems such as forests [20,21] or parks [22] or conversely, studies evaluated the ecosystem services provided by surface water [23–25] or groundwater [26,27]. While these studies have supported the field of ES assessment, no study has explicitly focused on the case of drinking water sources.

To effectively apply the ecosystem service approach to drinking water systems, it is necessary to define the system, i.e., all available drinking water sources, including their boundaries and scales. A drinking water source typically comprises surface water, ground-water, a combination of these, seawater, or rainwater (see Table 2). Other drinking water sources are also commonly used, e.g., water reclamation of wastewater but to integrate ES, we limit our attention to natural water sources.

**Table 2.** Overview of types of natural drinking water sources. In what manner the water is obtained from each source indicates the system boundaries. The list is exemplary and not exhaustive.

| Type of Natural Drinking Water Source | How Drinking Water Is Obtained | Examples of System Boundaries |
|---|---|---|
| Surface water | Pumping from river, reservoir, lake, canal | Freshwater body, recharge area, discharge area |
| Groundwater | Pumping from aquifer | Groundwater body, recharge area, discharge area, unsaturated zone |
| Combination of surface water and groundwater | Managed aquifer recharge | Groundwater body and freshwater body as well as their respective recharge area and discharge area, unsaturated zone |
| Seawater | Desalination | Seabed, water column, beaches, polder |
| Rainwater | Rainwater harvesting | Area for rainwater collection |

The cascade model from CICES can be applied to the water system (Figure 1). The natural conditions define essential characteristics of the water body (e.g., properties such as grain size and natural recharge). Properties of the water system determine the potential of drinking water sources to deliver services (e.g., potential abstraction rate). These services generate benefits for humans when they are used, and these benefits can be valued by, for example, assessing the associated economic value. Use typically entails the addition of some human-made or human-related inputs such as pipes for distributing drinking water to consumers or travel for being able to enjoy the beauty of a lake. Since services are a part of the natural system, they connect to the natural structures, processes, and functions that generate them.

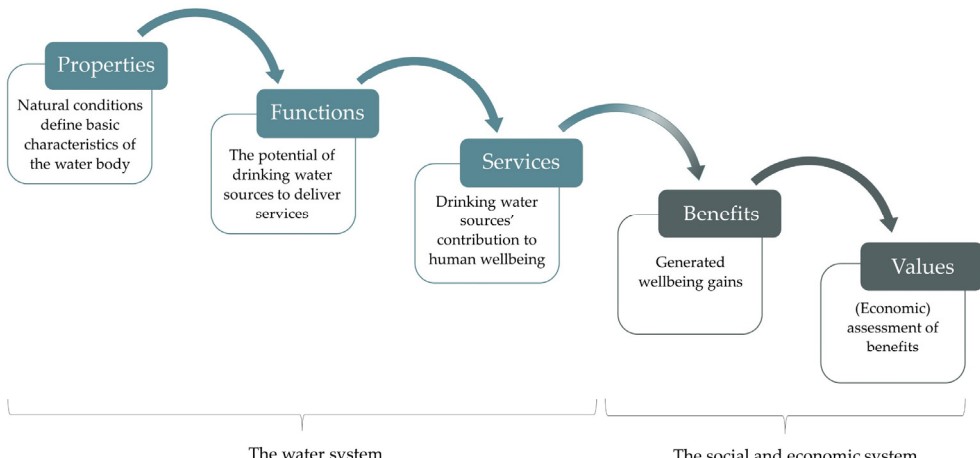

**Figure 1.** The CICES cascade model based on [5] applied to water systems.

Only a selection of WSS can be found in CICES. This selection includes abiotic services and biotic ecosystem services provided by a drinking water source (see Figure 2). However, the selection is not complete since more services related to drinking water sources can be present under certain conditions. To account also for these services, we developed the concept of water system services.

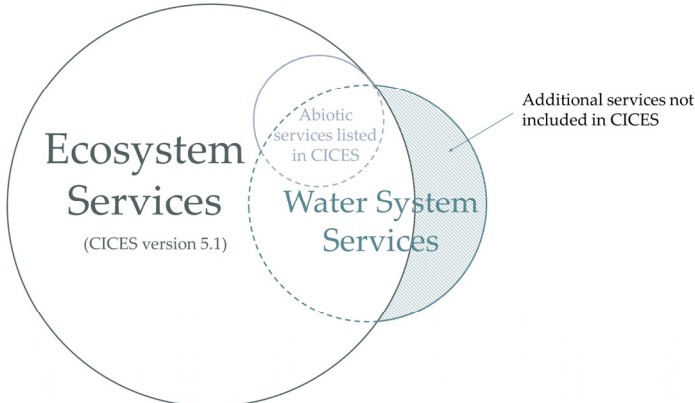

**Figure 2.** The relation between ecosystem services and water system services. Most water system services are ecosystem services, but additional services not included in CICES may exist depending on the type of drinking water source. Additional services can be biotic or abiotic.

The term water system services (WSS) was selected as the most suitable term to refer to the provision of water in sufficient quantity and quality for multiple uses. It was identified from a literature search on different terminology options, including *water services*, *water system services*, *water ecosystem services*, *hydrologic ecosystem services*, *water resource services*, *water source services*, and *water body services*. The search was directed by which terminology is the most suited for a drinking water source delimitation while avoiding confusion and overlapping with connotations in other disciplines. The terms were searched in Google Scholar and Scopus, and the results were examined on their publication date, sorted into specific disciplines, and the publications were then screened for the terms' meaning. For an overview of the terminology options, see Table S1 in the Supplementary Materials.

The concept of WSS builds on previous definitions of ES. Initially, ES were primarily understood as the benefits humans derive from a functioning ecosystem. More recent definitions make a clearer distinction between services and benefits [9]. Boyd and Banzhaf's refer to (final) ecosystem services as "*components of nature, directly enjoyed, consumed, or used to yield human well-being*" [28], while Fisher et al. [14] define them as "*the aspects of ecosystems utilized (actively or passively) to produce human well-being*". In line with these more recent

definitions of ES, we describe water system services as *the aspects of drinking water sources utilized to produce human well-being.*

## 3. Materials and Methods

In this section, we present the process for integrating ecosystem services into a risk assessment for drinking water protection, including the development of the region-specific list of WSS (Section 3.1), how it is applied to identify and quantify services (Section 3.2), and how it is integrated into a risk assessment (Section 3.3). We use the case study in Sweden to illustrate the process. A brief description of the case study site is provided in Section 3.4 (further details are available in Supplementary Material Table S4).

### 3.1. Development of a Region-Specific List of Water System Services

The process starts with the development of a region-specific list of WSS. The primary aim of the WSS list is to help users identify which services are provided by a drinking water source. The development of the list consists of five main steps. First, a pre-process is performed to determine the system's boundaries (i.e., the boundaries of drinking water sources). Second, the CICES service classes are reviewed and revised in several iterating steps. Third, for every WSS, at least one example is identified. Researchers (step four) and practitioners (step five) review the preliminary WSS list until they reach an agreement on a final version. In Figure 3, the procedure for compiling the WSS list is presented, and each step is further described in the following sections.

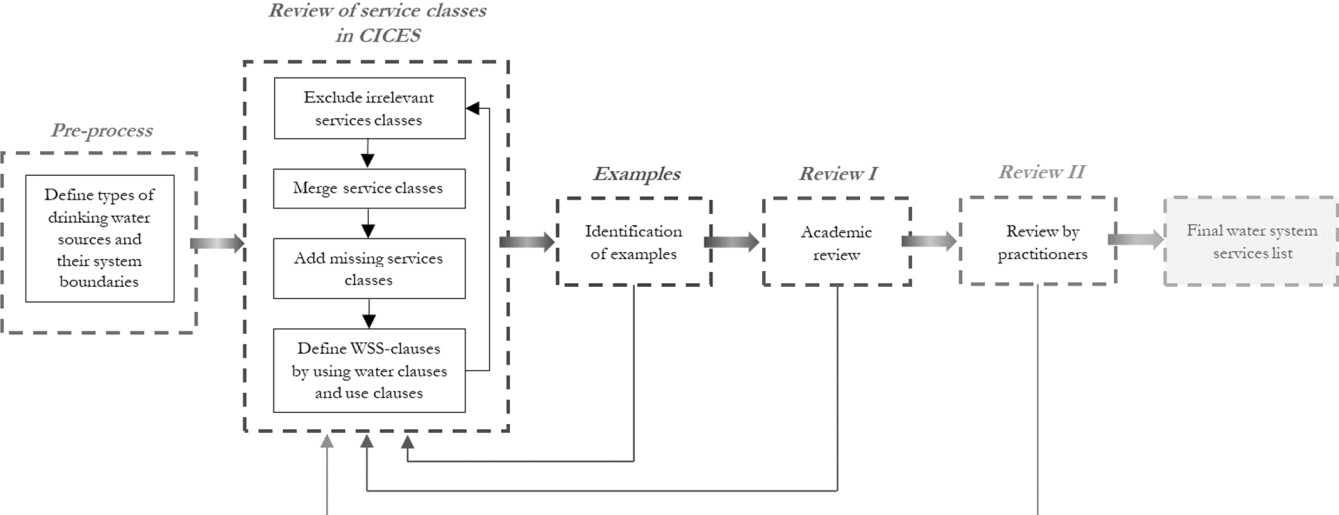

**Figure 3.** Applied procedure for compiling a list of water system services. The five steps to develop a list of WSS consist of the pre-process (step one), the review of service classes in CICES (step two), the identification of examples (step three), the review by academics (step four), and the review by practitioners (step five). The final output is a list of water system services.

### 3.1.1. Pre-Process (Step 1)

The first step in an ecosystem service assessment is to define the ecosystem's boundaries. Analogously, the scope of the water system has to be delimited, which requires defining types of drinking water sources relevant to the study. In our case study in Sweden, the drinking water source primarily comprises surface water, groundwater, or their combination (e.g., managed aquifer recharge). Hence, these types of water sources were the focus of the study. Furthermore, water sources feature unique spatial structures linked through water flow as parts of the hydrological cycle. When assessing the services provided by a drinking water source, the assessed area is delimited to surface freshwater bodies, groundwater aquifers, the interconnection of these as well as adjacent areas that are important for the hydrologic cycle and the function of water sources (e.g., recharge and discharge areas

and for groundwater aquifers—the unsaturated zone). This delimitation excludes other parts of the hydrological cycle and terrestrial systems, which are thus excluded from the analysis, but the support to those is acknowledged when assessing water system services.

3.1.2. Review of Service Classes in CICES (Step 2)

The review process starts from the complete CICES-list, version 5.1. We selected CICES as a base for our assessment as it is very detailed and contains the highest number of ecosystem service categories among existing systems. First, services considered irrelevant are excluded. Irrelevant services are those delivered by natural systems outside the scope of our defined drinking water source. Therefore, only groundwater sources, surface water sources, and combinations of these are considered. Marine ecosystem services classes and rainwater service classes are examples of excluded service classes.

Second, service classes are merged. A common reason for merging is similarities among biotic and abiotic classes in CICES. Care should be taken to keep a reference to CICES class codes, ensuring that the origin of merged classes is transparent. Third, additional service classes not listed in CICES may need to be added based on the specific features of the water systems. These classes are identified through a systematic search of the literature. For our assessment, we used the search engine Scopus to screen for relevant services mentioned in ecosystem services articles. Additional information regarding the literature search and the WSS identified is provided in Supplementary Material Table S2. Examples of excluded, merged, and added service classes are presented in Table 3.

**Table 3.** Examples of excluded, merged, and added service classes in the water system services list for Sweden. The code refers to the CICES class code.

|  | **Description of Service Classes** |
|---|---|
| *Excluded service class* | Coastal and marine water used as an energy source (code: 4.2.1.4) |
| *Merged service classes* | Pest control incl. invasive species (code: 2.2.3.1) and disease control (code: 2.2.3.2) |
| *Added service class* | Water as a means for transportation |

To maintain the same structure and way of defining services as in CICES, all water system service classes are defined using *water clauses* and *use clauses*. In CICES, every service class is defined by an ecological clause describing the biophysical output and a use clause describing its contribution to a use or benefit [9]. The ecological clauses and use clauses of WSS are defined following the CICES structure (see example in Table 4).

**Table 4.** Comparison of clause descriptions for service class erosion control (code: 2.2.1.1) in CICES and analogously in the water system services list for Sweden. The *water clause* corresponds to the *ecological clause* in CICES. The term *use clause* remains unchanged.

|  | **Ecological Clause** | **Use Clause** |
|---|---|---|
| *in CICES* | The reduction in the loss of material by virtue of the stabilizing effects of the presence of plants and animals . . . | . . . that mitigates or prevents potential damage to human use of the environment or human health and safety. |
|  | **Water Clause** | **Use Clause** |
| *in WSS* | The regulation in the loss of material, by virtue of the characteristics of aquatic ecosystems or by abiotic water characteristics, . . . | . . . that can protect people from erosion and mitigates or prevents potential erosion damage to human use, health, or safety. |

3.1.3. Identification of Examples (Step 3)

Examples of every service class are identified to ensure that only relevant services are included in the WSS list and that end-users have a contextual application of the service classes. In the process, first, all examples listed in CICES are reviewed to evaluate their

relevance to the case at hand. Second, at least one example is identified for each service class to ensure that the service classes are independent and non-repetitive. If the same example applies to several classes, the classes need to be merged or the service class description refined. If no relevant example can be identified, the service class should be excluded from the list.

3.1.4. Review Process (Steps 4 and 5)

The review process is two-tiered. The preliminary list from step 3 is first reviewed by a group of researchers (Step 4), followed by a group of potential end-users (Step 5).

The review by researchers (Step 4) is to ensure that all service classes on the list are clearly described and uniformly interpreted by researchers with different expertise and also to provide additional examples for the WSS selected. For the case of Sweden, the list was reviewed by a panel of five researchers (including the last four authors of this article and an additional member of the project team). In the process, the participants received the complete CICES-list, including merged classes, with a remark yes/no (for inclusion/exclusion, respectively) for each service class. Participants were asked to agree/disagree with the decisions. During two follow-up meetings, disagreements were discussed until a consensus was reached. In the last step, the experts were asked to add additional examples for each service class.

The review by potential end-users (Step 5) seeks to ensure that all service classes on the list are clearly described and uniformly interpreted by different types of users. The review process is also used to collect additional examples for the WSS on the list. In the Swedish case, we engaged a group of nine persons including representatives of national agencies, water utilities, consultancies, and authorities responsible for source water protection. Due to COVID-19 restrictions, the meeting was held online. During the event, the latest version of the list (previously circulated among the group) was discussed by the participants to expose ambiguities and inconsistencies. The process was concluded when a consensus was reached among the participants.

*3.2. Identification and Quantification of Water System Services for a Specific Site*

To identify the WSS present in a specific study site we employ a checklist approach. In the process, each service on the WSS list is evaluated based on remote sensing data and/or site visits to determine its presence. Instances of absence are illustrated and justified. The spatial extent of the assessed area includes the aquifer, the unsaturated zone, and the discharge area. A detailed overview of this process and data sources utilized in the Swedish case study can be found in Supplementary Material Table S4.

To quantify the WSS present in the area, we estimate their supply rate, and supply–demand ratio by employing the approach developed by [29]. The supply of WSS is expressed using a suitable indicator, e.g., $m^3/d$, number of wells, or visitors. The supply rate indicates how much of the maximum potential of a service is supplied at present, estimated as

$$\text{supply rate} = \frac{\text{supply}_{\text{actual}}}{\text{supply}_{\text{max}}}. \tag{1}$$

The supply–demand ratio illustrates the relationship between human demand and the supply of a WSS. The ratio might be positive (indicating a surplus of delivery), negative (indicating a deficit), or zero (supply and demand are in balance). Following [29], the relationship is defined as

$$\text{supply} - \text{demand ratio} = \frac{\text{supply}_{\text{actual}} - \text{demand}_{\text{human}}}{(\text{supply}_{\text{max}} + \text{demand}_{\text{max}}) \cdot \frac{1}{2}} \tag{2}$$

To provide useful decision support, the actual supply, the maximum supply, the maximum demand, and the actual human demand should be quantified when the WSS list is applied and when suitable indicators and data are available. The services and



their respective supply and demand indicators should be reviewed by experts and/or stakeholders who know the area.

### 3.3. Integration of Water System Services into a Risk Assessment

The risk assessment is typically divided into three key steps: hazard identification, risk estimation, and identification and evaluation of water protection measures [30]. The process of integrating WSS into the risk assessment for drinking water sources takes place in two of the three key steps, namely risk estimation, and identification and evaluation of water protection measures. In the hazard assessment, only hazards towards the drinking water source are identified.

### 3.3.1. Hazard Identification

A hazard identification aims to map and describe potential sources of risks to, in this case, a groundwater aquifer used as a water source. The applied approach follows general risk assessment guidelines for drinking water protection, starting with identifying hazards within the groundwater catchment area. However, the hazard assessment focuses only on hazards towards the drinking water service, and the spatial extent for the hazard identification includes only the area contributing to the drinking water well. Therefore, the area of the hazard assessment is incongruent with the area of the WSS assessment. The extent of the hazard assessment area was based on maps from the Geological Survey of Sweden [31].

To identify relevant hazards, the TECHNEAU-checklist was used as guidance. In 2008, Beuken et al. [32] developed and published the TECHNEAU hazard database, an extensive checklist of potential hazards in drinking water supply systems from source to tap. The hazard database is divided into 12 subsystems. In this study, the two subsystems *2. Groundwater catchment* and *5. Groundwater abstraction and transport* were used. The hazards listed in the TECHNEAU database were screened if they were present in the studied area, and if found positive, a detailed description of the hazard sources and the specific threat was documented. The screening was conducted remotely with GIS software, land use and land registry data, and previous reports from consultancies. A list of assessed hazards can be found in Supplementary Material Table S5. Analogously to the WSS assessment, the inventoried hazards were reviewed by experts and stakeholders who know the area.

### 3.3.2. Risk Estimation

Commonly, risk is described as an event that may occur and cause harm to a system that we aim to protect [33]. The WHO suggests using a risk matrix and calculating a risk priority number as part of developing a water safety plan [2]. A similar approach was adapted to fit both hazards and WSS. A risk priority number is calculated to illustrate the risk posed by each hazard and the overall risk the WSS are exposed to.

The risk ($R$) posed by a hazard ($i$) on a WSS ($j$) is here defined as

$$R_{ij} = l_i \cdot v_i \cdot c_{ji} \tag{3}$$

where $l_i$ is the likelihood of the hazard source ($i$) causing a discharge of a hazardous substance or, in other ways posing a potential threat to the water source, $v_i$ is the vulnerability of the water source concerning hazard $i$, and $c_{ji}$ is the consequence severity to WSS $j$ due to hazard $i$. The likelihood ($l$) and the vulnerability ($v$) are considered independent of the WSS, and thus only one likelihood and one vulnerability are defined for each hazard.

All variables ($l$, $v$, and $c$) are defined using integer rating scales from 0 or 1 to 5 according to Table 5. The likelihood, vulnerability, and consequence severity descriptions were adapted from the WHO's Water Safety Plan manual [34].

**Table 5.** Description of likelihood, vulnerability, and consequence severity classes and their respective scores.

| | Variable | Score | Description |
|---|---|---|---|
| *Likelihood (l)* | Most unlikely | 1 | Very uncommon event—probably will never occur |
| | Unlikely | 2 | The event may not occur |
| | Foreseeable | 3 | The event could occur |
| | Likely | 4 | The event has happened before and can probably occur again |
| | Almost certain | 5 | A very common event, occurs regularly |
| *Vulnerability (v)* | Insignificant | 1 | The water source is barely vulnerable to a hazardous event |
| | Low | 2 | The water source has a very good ability to withstand the effects of the hazardous event |
| | Moderate | 3 | The water source has a good ability to withstand the effects of the hazardous event |
| | High | 4 | The water source has very little ability to withstand the effects of the hazardous event |
| | Extreme | 5 | The water source cannot withstand the effects of the hazardous event |
| *Consequence severity (c)* | No consequences | 0 | WSS will not be affected if the hazardous event occurs |
| | Insignificant | 1 | Insignificant potential to cause harm to WSS |
| | Minor | 2 | Potential to cause minor discomfort to WSS |
| | Moderate | 3 | Potential to cause a moderate impact on WSS (no long-term consequences) |
| | Major | 4 | Potential to cause a major negative impact on WSS (incl. long-term consequences) |
| | Catastrophic | 5 | Potential to cause a catastrophic negative impact on WSS (incl. long-term consequences) |

The total risk posed by a hazard ($i$) is calculated as the sum of the risk for all WSS ($m$).

$$R_i = \sum_{j=1}^{m} R_{ij} = \sum_{j=1}^{m} l_i \cdot v_i \cdot c_{ji} \tag{4}$$

Furthermore, the total risk a WSS ($j$) is exposed to is calculated based on all hazards ($n$), as

$$R_j = \sum_{i=1}^{n} R_{ij} = \sum_{i=1}^{n} l_i \cdot v_i \cdot c_{ji} \tag{5}$$

Four categories of risk levels are defined to evaluate estimated risk priority numbers and help identify the most severe risks. This is performed for $R_{ij}$, i.e., the risk posed by one hazard to one WSS. The maximum possible risk value is 125 and the risk categories are: 0 = no risk, 1–40 = low risk, 41–100 = medium to high risk, >100 = extremely high risk.

3.3.3. Identification and Evaluation of Water Protection Measures

Based on the results from the risk assessment, the need for measures aimed at reducing the risk is identified. The most important hazards to be targeted can be identified based on the services that are exposed to the highest risk. Here we use a set of potential water protection measures to illustrate the process.

The effect of protective measures can be estimated based on either the overall risk reduction or risk reduction towards specific services. The risk reduction by a protective measure ($k$) is here referred to as the benefit ($u$) and the risk reduction a measure provides for a specific hazard, and WSS is calculated as

$$\Delta R_{ij} = R_{0ij} - R_{kij} \tag{6}$$

where $R_{0ij}$ is the initial risk level prior to any protective measure, and $R_{kij}$ is the estimated residual risk after measure $k$ ($k > 0$) has been implemented. The overall benefit of a protective measure can be calculated as

$$u_k = \sum_{i=1}^{n} \sum_{j=1}^{m} \Delta R_{ij} \tag{7}$$

Furthermore, the total risk reduction for a certain WSS by a protective measure is calculated as

$$u_{kj} = \sum_{i=1}^{n} \Delta R_{ij} \tag{8}$$

### 3.4. Case Study Site–Skallsjö

The case study site is situated in the locality of Floda in Lerum Municipality, Västra Götaland County in South-West Sweden, approximately 30 km northeast of the city of Gothenburg (see Figure 4).

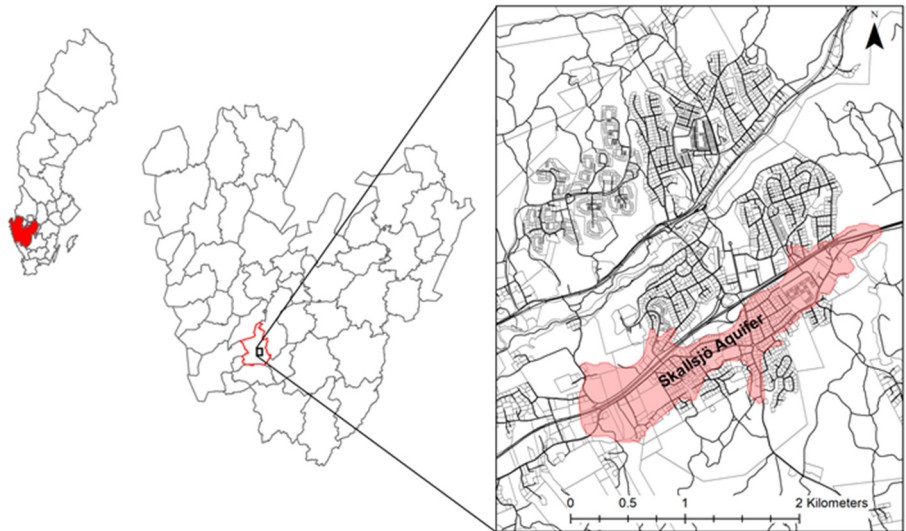

**Figure 4.** Location of the case study area. The Skallsjö aquifer is located in Lerum Municipality in Västra Götaland County in Sweden.

The Skallsjö aquifer was chosen as a case study site because it is exemplary for a Swedish groundwater body where the three factors of typical hydrogeology (glaciofluvial deposit), representative land cover (semi-rural with urban fabric and some industrial activities), and a small daily pumping rate (740 m$^3$/day) are combined. Furthermore, the aquifer has good data availability as the site has been investigated by the Geological Survey of Sweden and consultancy companies during previous projects.

### 4. Results

#### 4.1. Water System Services List

Figure 5 shows the transformation of the ES framework CICES v.5.1 into the water system services list. Compared to the initial amount of service classes in CICES, the WSS list resulted in fewer service classes, fewer groups, and fewer divisions. The complete developed list of provisioning, regulating, and cultural WSS is presented in Appendix A (see Table A1, Table A2, and Table A3, respectively). In addition to the identified services, the related CICES codes, water clauses and use clauses, examples, and the development from CICES to WSS are presented in Supplementary Material Table S3.

The identified water system services are tailored for Swedish drinking water sources. The list is applicable for an assessment in Sweden or a region with comparable drinking water sources. However, for regions with significantly different conditions, we recommend adapting the list according to the steps presented in Section 3.1.

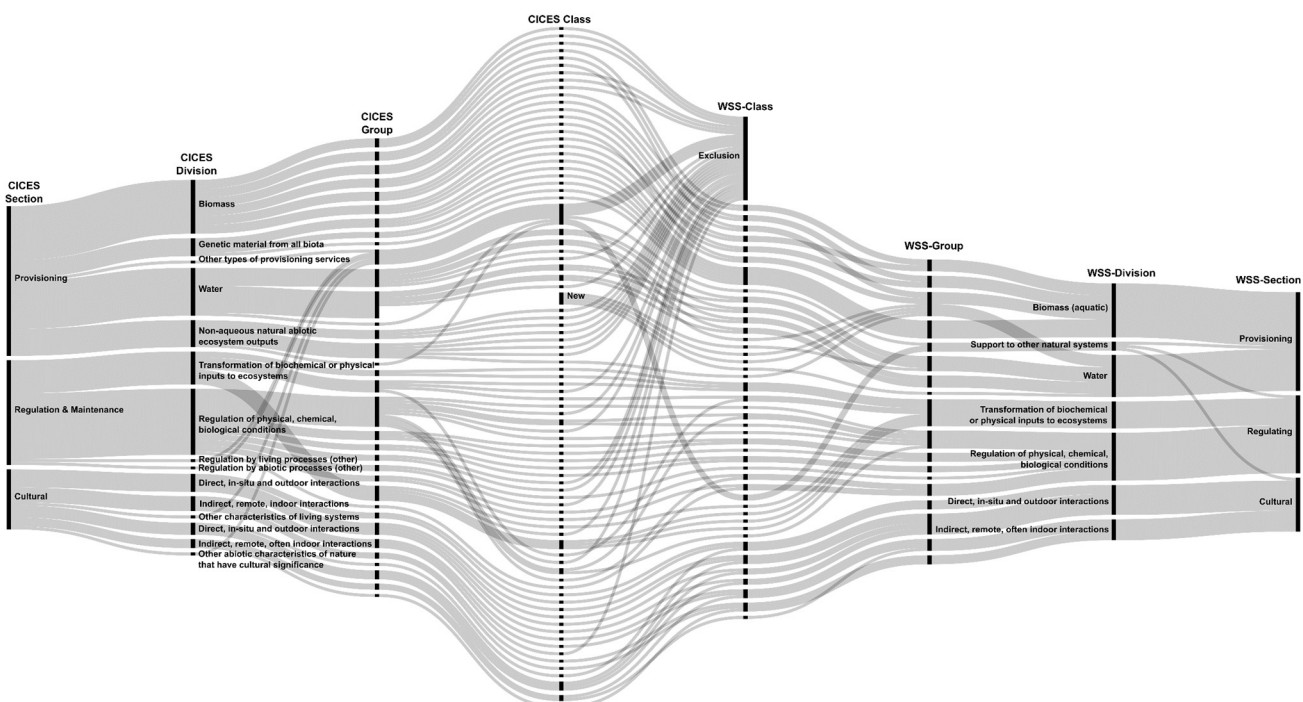

**Figure 5.** Transformation of the ecosystem service framework CICES v.5.1 into the water system services list. The names of CICES groups, CICES classes, and WSS classes are not displayed to provide better readability. For the same reason, the abiotic and biotic CICES sections have been merged. A detailed description of all classes can be found in Supplementary Material Table S3.

### *4.2. Case Study Results*

The practical applicability of the water system service list was tested by applying it to a case study and using the results as input to a risk assessment for drinking water protection. The use of a case study aimed to gain general insights on data availability and the degree of difficulty when risk assessments with WSS are applied to a real-world case.

#### 4.2.1. Identified Water System Services

Using the developed list, eleven water system services were identified for the Skallsjö aquifer. Services were found in each category, i.e., provisioning, regulating, and cultural services. In Table 6, the identified services are quantified regarding their actual supply, and it is illustrated where the services are provided in the catchment. A more detailed description of supplied services, including the actual supply, the maximum supply, the maximum demand, and the actual human demand, are listed in Table S4 in the Supplementary Material.

#### 4.2.2. Identified Hazards

Based on the TECHNEAU hazard database, land use data, and site visits, hazards were identified within the area of contribution defined for the municipal drinking water wells. We identified twelve hazards that pose a risk to the municipal drinking water supply (see Supplementary Material Table S5 for a detailed description). The hazards and their locations are presented in Figure 6. Most of the hazards can be allocated to human activities within the studied area. To illustrate a hazard that cannot be mitigated by local measures, reduced groundwater recharge due to climate change is included as a potential, although unlikely, hazard and is referred to as *less precipitation*. Furthermore, the hazards include potential contamination sources (e.g., discharge in case of a road accident or the spreading of pesticides) that may affect the water quality and hazards that may affect the natural groundwater recharge (e.g., due to impermeabilization). *Heat pumps* and *household wells* are

hazards that exist due to the use of services which demonstrates that some services can also provoke hazards.

**Table 6.** Identified water system services from the Skallsjö aquifer.

| WSS Division and Class | | Description (Quantification) | Location |
|---|---|---|---|
| **Provisioning services** | Water supply for humans | Municipal drinking water supply (740 m$^3$/day) | 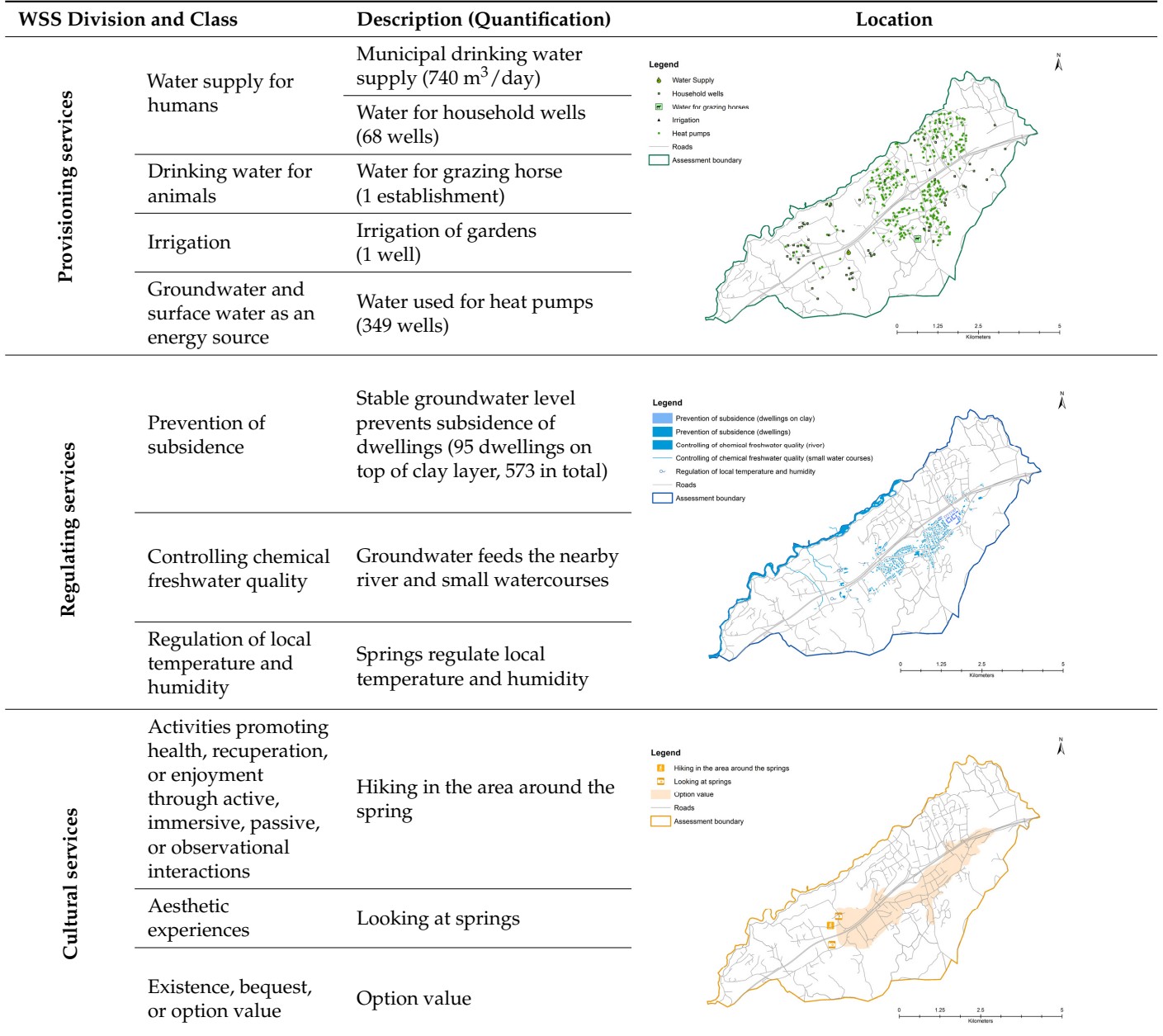 |
| | | Water for household wells (68 wells) | |
| | Drinking water for animals | Water for grazing horse (1 establishment) | |
| | Irrigation | Irrigation of gardens (1 well) | |
| | Groundwater and surface water as an energy source | Water used for heat pumps (349 wells) | |
| **Regulating services** | Prevention of subsidence | Stable groundwater level prevents subsidence of dwellings (95 dwellings on top of clay layer, 573 in total) | |
| | Controlling chemical freshwater quality | Groundwater feeds the nearby river and small watercourses | |
| | Regulation of local temperature and humidity | Springs regulate local temperature and humidity | |
| **Cultural services** | Activities promoting health, recuperation, or enjoyment through active, immersive, passive, or observational interactions | Hiking in the area around the spring | |
| | Aesthetic experiences | Looking at springs | |
| | Existence, bequest, or option value | Option value | |

### 4.2.3. Risk Estimation

The Skallsjö aquifer delivers various WSS but is simultaneously pressured by twelve different hazard sources. The assessment matrix in Figure 7 contrasts the hazard sources and their potential impact on each water system service. A risk score for each hazard-service pair was calculated based on the equations and tables in Sections 3.3.2 and 3.3.3. A water protection area exists for the Skallsjö aquifer in which the land use is already restricted to manage the risks. Therefore, existing restrictions have been considered when estimating the risk, which is why most hazards pose a low or medium risk to the drinking water service.

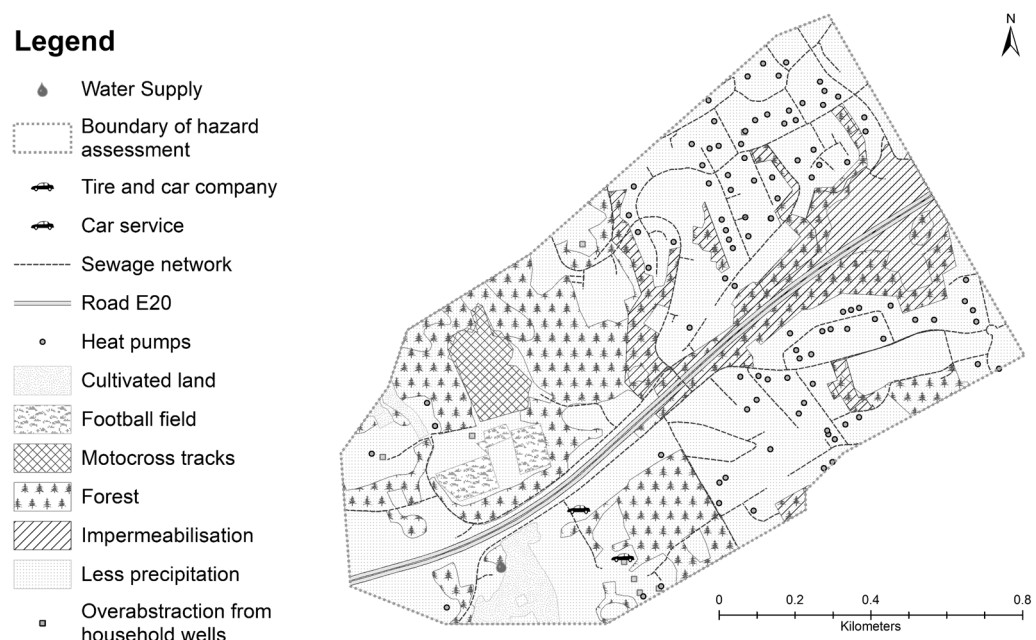

**Figure 6.** Identified hazard sources that pose a risk to the municipal drinking water supply.

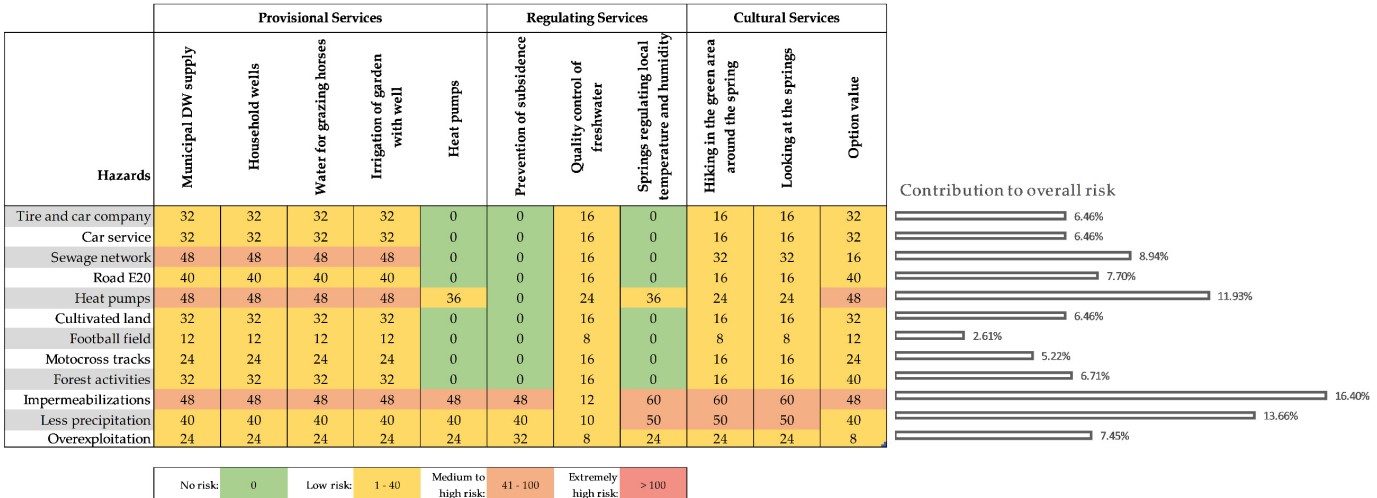

**Figure 7.** Assessment matrix contrasting hazard sources and their impact on water system services. Numbers from 0 (minimum risk score) to 125 (maximum risk score possible) in the matrix represent the calculated risk score for each hazard-service pair. The bar chart on the right indicates the contribution of each hazard source to the overall risk.

The potential hazard of *impermeabilizations* contributes most to the overall risk, i.e., the risk posed to all services, followed by *less precipitation*. In general, hazard sources that affect the source water quantity will have a more significant impact across all services than hazard sources that only affect the water quality. If there is no water, no service can be delivered, whereas some services can still be provided even if the water quality deteriorates.

The provisioning services (excl. *heat pumps*) and the cultural service *option value* are similarly dependent on water quantity and quality and are exposed to almost the same risks. However, the heat pump service and parts of the regulating services are only dependent on the water quantity and thus not affected by several hazards. A detailed itemization for each risk score is available in Table S6 in the Supplementary Material.

4.2.4. Output for Decision Support

Based on the assessment matrix (Figure 7), there are two ways to select hazard sources for mitigation. We can either address hazard sources that are the most significant contributors to the overall risk (a long bar in the bar chart indicates a high contribution to the overall risk), or we can target hazard sources that contribute most to a service that is especially worth protecting. For the latter option, the highest scores within a column are identified and suggest the hazards for mitigation. In this study, the assessment aims at protecting the drinking water service but at the same time clearly illustrates positive and negative effects on other services.

In the case study, *impermeabilization*, *the sewage network*, and *heat pumps* pose the highest risk to the drinking water service but *impermeabilization* is the largest contributor to the overall risk with 16.4%. The second-largest overall risk is *less precipitation*. However, the municipality or the water utilities cannot mitigate this hazard as mitigating climate change is outside their action radius. Therefore, we use measures directed at the *heat pump* and *overexploitation* as examples to decrease the risk posed to the drinking water service and to reduce the overall risk. These two hazards contribute considerably to the overall risk, and once mitigation measures are implemented, there is an immediate risk reduction. The analyzed measures are: (1) prohibit the use of *ground source heat pumps* within the boundaries of the hazard assessment to avoid contamination; (2) prohibit the use of *household wells* within the recharge and protection area to avoid overexploitation. Implementing these two mitigation measures was estimated to reduce the overall risk by nearly 20%. Figure 8A shows the comparison between the relative reduction in risk versus the total reduction in risk score with the implementation of the mitigation measures for each water system service.

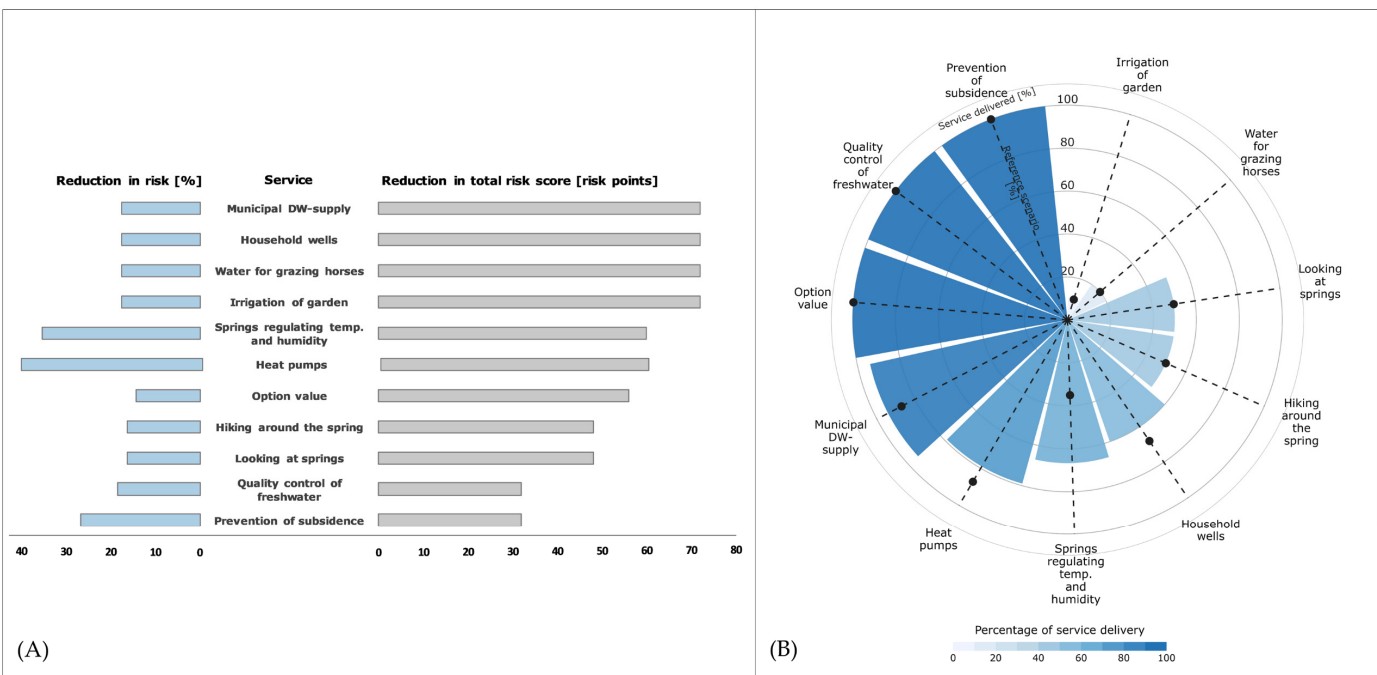

(A)

(B)

**Figure 8.** (**A**) Comparison between the relative reduction in risk versus the total reduction in risk score with the implementation of the mitigation measures (restriction of ground source heat pumps and restriction of household wells) for each water system service. (**B**) The visualization shows the estimated water system services delivered by the Skallsjö aquifer once the mitigation measures are established. The black dots refer to the service supply of the reference scenario (situation today). The percentage of service delivery states the actual supply in the scenario, 100% of delivered service equals the maximum possible supply of each service.

There is a need to illustrate the negative consequences (trade-offs) as well as the positive consequences (synergies) of mitigation measures when mitigating risks towards drinking water sources and implementing water protection measures. Synergies occur when mitigation measures in pursuit of improving one service produce an improvement in other services. In contrast, trade-offs appear when improving one service leads to the deterioration of another service. Figure 8B illustrates the estimated change in water system services delivery by the Skallsjö aquifer if the mitigation measures (restriction of ground source heat pumps and restriction of household wells) are established. The black dots refer to the service supply of the reference scenario (situation today), and the petals show the estimated service supply if mitigation measures are implemented. Due to the restrictions, the services of *ground source heat pumps* and *household wells* declined, whereas the *municipal drinking water supply* and the *springs regulating temperature and humidity* improved. Since there are numerous options for reducing the overall risk and what to mitigate, the illustration of trade-offs and synergies between services builds a basis for decision-making and communication with stakeholders.

## 5. Discussion

### 5.1. Insights from the Case Study Application

Applying our approach to the Skallsjö aquifer demonstrates that integrating ecosystem services into a risk assessment is feasible. This approach provides valuable information for identifying and mitigating risks toward drinking water sources. We found that the aquifer provides eleven water system services but is simultaneously pressured by twelve hazards.

There is a need for straightforward, cost-effective methods when integrating ecosystem services into water management [8], and the development of an operable list constituted one of the principal objectives of this study. It is possible to conduct an ecosystem service assessment using a complete ecosystem service catalog for every water source (as demonstrated by [24,27]) and identify the services without the WSS list. However, this results in a time-consuming assessment process. Compared to general ES lists, the WSS list promotes uncomplicated identification of services and can be easily applied by practitioners without extensive knowledge of ecosystem service research.

The risk assessment matrix provides a clear overview of the interplay between hazards and services. It demonstrates that some services are much less affected by hazards than others, whereas some hazards pose a risk to almost all services. It was also shown that an identified WSS (e.g., ground source heat pumps) can simultaneously be a service and give rise to a hazard and thus pose a risk to other services. The results from the case study illustrate that most hazard sources pose a relatively low risk due to the current regulations and existing water protection measures. The hazard *less precipitation* poses the second-largest overall risk but is not manageable by the water utility, which also reflects the generally low risk as there are no urgent contamination sources.

We want to stress that this assessment presents the current conditions at the Skallsjö aquifer. However, if the area undergoes a development, additional hazards will likely appear and pose new risks to the aquifer, at which point the assessment should be updated.

### 5.2. Towards Ecosystem Services-Based Decisions in Drinking Water Protection

If water sources stay uncontaminated and services are safeguarded, expensive treatment systems are avoided [4,27], justifying preventive measures to protect the water source. Incorporating ecosystem services into water safety plans adds to the WHO's approach by making it possible to motivate protective measures and perform comprehensive assessments of water sources more efficiently.

Even though there are benefits in avoiding expensive treatments when source water is protected, drinking water protection always bears a cost, and there is a need to illustrate the negative consequences (trade-offs) and the positive consequences (synergies) of mitigation measures. Highlighting the consequences raises awareness of drinking water protection and aids in implementing protection measures [27]. WSS support human well-being, and

making these services visible contributes to a higher appreciation of the drinking water source. The illustration of WSS (as in Figure 8B) paves the ground for utilizing synergies and limiting trade-offs.

Having a clear and transparent representation of services, hazards, and risk mitigation consequences improves decision support. For example, a large set of hazards in a drinking water catchment is common but identifying services aside from the drinking water service adds another dimension to the assessment. Either protective measures can be determined based solely on the drinking water service or based on the effect they have on all services, or we can identify measures based on the risk posed to all services and, in this case, also assess the overall effect.

Following the rationale of IWRM [35], incorporating ecosystem services assessments is the latest development water management has undergone [36]. However, due to its complexity, ES faced significant critique for not being compatible with governance and management of water resources [37]. The list of WSS provides an instrument to close this implementation gap.

*5.3. Limitations and Recommendations*

Ecosystems in general, and water sources in particular, are dynamic structures that naturally vary over space and time and over short time scales [38]. This assessment only considered WSS statically and locally and disregarded services at the regional or global scale, such as carbon sequestration or global climate regulation. Spill-over effects from the local to the global scale are often not sufficiently explored [39], leading to underestimating the delivered services.

In this study, a risk-scoring method was used when assessing the overall risk, similar to the approach suggested by the WHO as part of a Water Safety Plan. However, it is possible to advance the risk estimation by other risk assessment methods tailoring assessments to a specific study purpose [40]. Furthermore, all services in the assessment were regarded as equally important, and no weight was assigned to them. It can, for example, be argued that the service *irrigation of gardens* is not as vital to society as the service *ground source heat pumps* when prioritizing between different mitigation measures. One way to consider the weighting of services is by using multi-criteria decision analysis (MCDA). Marttunen et al. [41] present an overview of how MCDA can complement ecosystem service assessments in water management. Another possibility is to consider how protective measures affect the benefits provided by water system services and estimate the change in economic value to society, according to the cascade model in Figure 1. Grizzetti et al. [36] recommend an economic valuation of ecosystem services after the biophysical assessment of ES. The combined WSS and risk assessment results can serve as a key input to cost–benefit analysis (CBA) to guide the social profitability of protective measures [42]. A CBA will require information on the economic value of drinking water, the remaining water system services, and the costs of implementing protective measures.

This study is a first attempt to integrate ecosystem services into a risk assessment for drinking water protection. The application of water system services is part of an ongoing research project [43], and with an increasing number of applications, we expect to get more insights into the procedure, data, and end-user preferences. Future research will include further development on using WSS and risk assessment results as input to a cost–benefit analysis and collecting necessary valuation data to consider the drinking water service's economic value appropriately.

## 6. Conclusions

Drinking water protection always bears a cost, and there is a need to transparently illustrate the negative consequences (trade-offs) and the positive consequences (synergies) of mitigation measures. Conventional risk assessments, as suggested by the WHO, do not address these consequences. When ES are included in a risk assessment, trade-offs and synergies of protection measures can be illustrated and quantified. However, both the ES

assessment and the risk assessment method have to be operationalized to be able to integrate them. The approach we presented here suggests the development of a region-specific WSS list and an extension of the risk assessment's scope to support operationalization and integration. Applying the approach to a case study demonstrates its practical feasibility for drinking water sources.

The concept of WSS matches the desire to expand the focus of spatial planning beyond the drinking water protection when implementing risk mitigation measures. Planners who acknowledge and implement a broader scope of services into the planning for drinking water protection will facilitate communication and negotiation with stakeholders and increase the social acceptability of mitigation measures.

**Supplementary Materials:** The following are available online at https://www.mdpi.com/article/10.3390/w14081180/s1, Table S1: Overview of terminology options in different databases and the dominant use of those options in different disciplines. Table S2: Literature review on additional services. Table S3: Transformation from CICES to WSS. Table S4: Identification and Quantification of WSS in the Case Study Area. Table S5: Hazard Assessment. Table S6: Risk Assessment for Skallsjö.

**Author Contributions:** Writing—original draft preparation, N.G.; writing—review and editing, N.G., A.L., L.R., T.S. and J.N.; conceptualization, N.G., A.L. and T.S.; methodology—WSS list, N.G., A.L., T.S., J.W., L.R., H.N., J.N. and L.-O.L.; methodology—risk assessment, N.G. and A.L.; software, N.G.; validation, N.G., A.L. and L.-O.L.; formal analysis, N.G.; investigation, N.G.; data curation, N.G.; visualization, N.G.; supervision, A.L.; project administration, A.L.; funding acquisition, A.L., L.R., T.S. and J.N. All authors have read and agreed to the published version of the manuscript.

**Funding:** This research was funded by Formas, a Swedish research council for sustainable development (grant number 2018-00202).

**Informed Consent Statement:** Not applicable.

**Acknowledgments:** This research was performed within the DRICKS center for drinking water research coordinated by the Chalmers University of Technology. The presented work has been part of the WaterPlan project (www.waterplanproject.org, accessed on 28 March 2022), and the authors would like to thank the project's reference group and water utilities. We also want to thank Linus Hasselström, who participated in the academic review process and the general development of the WSS list for Sweden.

**Conflicts of Interest:** The authors declare no conflict of interest.

## Appendix A. Water System Services List for Sweden

**Table A1.** Provisioning services. The water clause and use clause define the class.

| Division | Group | Code Used in CICES v 5.1 | Class | Examples of Services | Water Clause | Use Clause |
|---|---|---|---|---|---|---|
| Biomass (aquatic) | Food | 1.1.2.1, 1.1.4.1 | Cultivated plants or animals | - Crayfish, char, eel, rainbow trout, steelhead trout, salmon | Nature's contribution to the growth of organisms in aquaculture . . . | . . . that can be harvested and used as raw material for the production of food |
| | | 1.1.5.1, 1.1.6.1 | Wild plants or animals | - Perch, pike, zander, rainbow trout, char, roe from fish<br>- Watercress | Parts of the standing biomass of non-cultivated aquatic organisms and their outputs . . . | . . . that can be harvested and used as raw material for the production of food |
| | Material | 1.1.2.2, 1.1.4.2 | Fibers and other materials from cultivated plants or animals | - Jewelry with fish scales as adornment | Nature's contribution to the growth of organisms in aquaculture . . . | . . . that can be harvested and used as raw material for non-nutritional purposes |
| | | 1.1.5.2, 1.1.6.2 | Fibers and other materials from wild plants or animals | - Jewelry with fish scales as adornment<br>- Reed for roofs and crafts or as food for animals | Parts of the standing biomass of non-cultivated aquatic organisms and their outputs . . . | . . . that can be harvested and used as raw material for non-nutritional purposes |
| | Energy | 1.1.2.3, 1.1.4.3 | Cultivated plants or animals as an energy source | - Reed canary grass for biofuel (fuel pellets)<br>- Biogas from aquaculture waste | Nature's contribution to the growth of organisms in aquaculture . . . | . . . that can be harvested and used as a source of energy |
| | | 1.1.5.3, 1.1.6.3 | Wild plants or animals as an energy source | - Wild reed for heating | Parts of the standing biomass of non-cultivated aquatic organisms and their outputs . . . | . . . that can be harvested and used as an energy source |
| | Genetic Material | 1.2.1.1, 1.2.1.2, 1.2.1.3, 1.2.2.1, 1.2.2.2, 1.2.2.3 | Genetic material from all organisms | - Wild animals that we can use for breeding<br>- Plants, fungi, or algae that we can use for breeding | Genetic material and information from aquatic organisms . . . | . . . that can be used to maintain, develop new varieties or establish a new population, or that can be used in gene synthesis |
| Water | Water for drinking | 4.2.1.1, 4.2.2.1 | Water supply for humans | - Municipal water supply<br>- Potable water in the public supply system<br>- Potable water using private wells<br>- Natural springs<br>- Mineral water | Surface water bodies or aquifers . . . | . . . that provide a source of drinking water supply for humans |

**Table A1.** *Cont.*

| Division | Group | Code Used in CICES v 5.1 | Class | Examples of Services | Water Clause | Use Clause |
|---|---|---|---|---|---|---|
| | | 4.2.1.1, 4.2.2.1 | Drinking water for animals | - Water source for wild animals<br>- Water source for livestock | Surface water bodies or aquifers . . . | . . . that provide a source of drinking water supply for animals |
| | | own description | Reserve water sources | - Potable water system if an existing source cannot be used<br>- Potable water system in the future (e.g., if the demand increases) | Surface water bodies or aquifers . . . | . . . that provide a source of reserve drinking water supply |
| | Water for non-drinking purpose | 4.2.1.2, 4.2.2.2 | Irrigation | - Land irrigated by controlled flooding<br>- Irrigated temporary grass land, cereals, potatoes, sugar beet, etc. | Surface water bodies or aquifers . . . | . . . that provide water which can be used for irrigation |
| | | 4.2.1.2, 4.2.2.2 | Water used as a material or other type of input into production and consumption | - Water for washing (in industrial processes)<br>- Water for sanitation<br>- Water for mortar<br>- Water used as a transport medium in heating or cooling systems, e.g., in industrial facilities<br>- Water as a material for fire extinguishing | Surface water bodies or aquifers . . . | . . . that provide water which can be used as other types of input into production and consumption |
| | Energy | 4.2.1.3 | Surface water in hydropower | - Potential and kinetic energy in water that can be used in hydropower to produce electricity | The flow of water on land . . . . | . . . that can be converted to electrical or mechanical energy |
| | | 4.3.2.5 | Geothermal energy | - Hot water and steam from (typically deep boreholes) that can be used for the production of electricity and heating | Hot water and steam from the subsurface of the earth . . . | . . . that can be used as an energy source |
| | | own description based on 4.2.2.3 | Groundwater and surface water as an energy source | - Groundwater source heat pump<br>- District cooling and heating | Surface water bodies or aquifers . . . | . . . that provide water at useful temperatures |
| | | own description | Water as storage of heat and coolness | - Seasonal storage of heat and coolness in aquifers, e.g., heating and cooling of facilities (Arlanda airport) | Surface water bodies or aquifers . . . | . . . that provide a source for storage of heat or coolness |
| | Water for transport | own description | Water as a means of transportation | - Water bodies as a waterway for ships<br>- Frozen surface water bodies as winter ways for vehicles | Surface water bodies or aquifers . . . | . . . that provide water which can be used as a mode of transport |
| Support to other natural systems | | | | | Surface water bodies or aquifers forming input to the functioning of other natural systems . . . | . . . that provide provisioning services |

**Table A2.** Regulating services. The water clause and use clause define the class.

| Division | Group | Code Used in CICES v 5.1 | Class | Examples of Services | Water Clause | Use Clause |
|---|---|---|---|---|---|---|
| Transformation of biochemical or physical inputs to ecosystems | Mediation of waste, toxic substances, and nuisances | 2.1.1.1, 2.1.1.2, 2.1.2.1 | Through living processes | - Denitrification<br>- Biofilm in infiltration ponds<br>- Biological degradation of organic substances (petroleum products, chlorinated solvents) | Transformation, fixing, and storage of an organic or inorganic substance, and reducing the impact of odors by aquatic organisms . . . | . . . that mitigate harmful effects or reduce the costs of disposal by other means |
| | | 5.1.1.1 | Through dilution | - Treated wastewater discharged from a wastewater treatment plant into a surface water body for an effluent dilution | The reduction in the concentration of organic or inorganic substances by mixing in freshwater, . . . | . . . that mitigates harmful effects or reduce the costs of disposal by other means |
| | | 5.1.1.3 | Through filtration | - Surface water bodies filter air pollution | Mediation through filtration of waste, toxins, and other nuisances, by chemical and physical processes of water, . . . | . . . that can protect people |
| | | 5.1.1.3 | Through sequestration | - Nutrient degradation (phosphorus capture) | Mediation through sequestration of waste, toxins, and other nuisances by chemical and physical water processes, . . . | . . . that can protect people |
| | | 5.1.1.3 | Through storage or accumulation | - Decommissioning of open-pit mines in order to prevent acid mine drainage (e.g., Udden)<br>- Natural sedimentation of pathogens in a surface water body | Mediation through storage or accumulation of waste, toxins, and other nuisances by chemical and physical water processes, . . . | . . . that can protect people |
| | | 2.1.2.3, 5.1.2.1 | Through other water-related mediation | - Visually covering the open pit (open-cast) with water and establishing an artificial lake, open pit (open-cast) mining lakes (e.g., Udden) | Other types of water-related mediation of environmental conditions . . . | . . . . that can reduce or mitigate nuisance to people |
| Regulation of physical, chemical, biological conditions | Regulation of baseline flows and extreme events | 2.2.1.1 | Erosion control | - Stable surface water level to e.g., prevent landslides in Göta Älv river<br>- Reed | The regulation in the loss of material, by virtue of the characteristics of aquatic ecosystems or by abiotic water characteristics, . . . | . . . that can protect people from erosion and mitigates or prevents potential erosion damage to human use, health, or safety |
| | | 2.2.1.3, 5.2.1.2 | Flood protection | - Surface water bodies receive excess water and provide flood protection | The regulation of water flows, by virtue of the characteristics of aquatic ecosystems or by abiotic water characteristics, . . . | . . . that can protect people from flooding and mitigates or prevents potential flooding damage to human use, health, or safety |

**Table A2.** *Cont.*

| Division | Group | Code Used in CICES v 5.1 | Class | Examples of Services | Water Clause | Use Clause |
|---|---|---|---|---|---|---|
| | | own description based on 5.2.1.2 | Prevention of subsidence | - Prevention of subsidence by maintaining a stable groundwater level | The regulation of water flows, by virtue of the characteristics of aquatic ecosystems or by abiotic water characteristics, . . . | . . . that can protect people from subsidence and mitigates or prevents potential subsidence damage to human use, health, or safety |
| | | own description based on 2.2.1.3 | Drought attenuation | - A surface water body that retains water and is able to release it slowly <br> - Groundwater leaves the subsurface via springs and wetlands | The regulation of water flows, by virtue of the characteristics of aquatic ecosystems or by abiotic water characteristics, . . . | . . . that can protect people from drought and mitigates or prevents potential drought damage to human use, health, or safety |
| | | 2.2.1.5 | Fire protection | - Rivers or other surface water bodies as a physical barrier against fires (fire protection belt) | The reduction in the incidence, intensity or speed of fire spread by virtue of the presence of aquatic organisms and the presence of water in the landscape, . . . | . . . that can protect people from fire and mitigates or prevents potential fire damage to human use, health, or safety |
| | Lifecycle maintenance, habitat and gene pool protection | 2.2.2.1, 2.2.2.2 | Pollination and spreading of seeds by water | - Seeds of aquatic (macrophytes) and non-aquatic plants that are dispersed by water <br> - Non-aquatic plants: e.g., water mint | The water-related dispersal of seeds and spores, and the fertilization of crops, by aquatic organisms, . . . | . . . that maintains or increases the abundance and/or diversity of organisms that are important to people in use or non-use terms |
| | | 2.2.2.3 | Maintaining populations and habitats | - The water body provides a natural habitat for species, e.g. (gravel areas for spawning sea trout) | The presence of ecological conditions (usually habitats) and abiotic conditions necessary for sustaining populations of aquatic organisms . . . | . . . that are important to people in use or non-use terms |
| | Pest and disease control | 2.2.3.1, 2.2.3.2 | Pest and disease control | - Regulation of pathogens and parasites by (aquatic) organisms, pH, and UV-light | The reduction, carried out by aquatic biological and water interactions or by the presence of water bodies, of the incidence of organisms . . . | . . . that prevent or reduce the output of food, material or energy, or their cultural importance |
| | Maintaining water conditions | 2.2.5.1 | Controlling the chemical quality of freshwater | - Managed aquifer recharge <br> - Living processes maintaining the already acceptable water quality | Maintenance of the chemical condition of freshwaters, by aquatic organisms or by abiotic water characteristics, . . . | . . . that enables human use, health, or safety |

**Table A2.** *Cont.*

| Division | Group | Code Used in CICES v 5.1 | Class | Examples of Services | Water Clause | Use Clause |
|---|---|---|---|---|---|---|
| | Atmospheric composition and conditions | 2.2.6.1, 5.2.1.3 | Regulation of global climate | - Lake sediments that accumulate organic matter are effective long-term sinks for carbon (specifically for boreal and northern lakes)<br>- Carbon sequestration in rivers (there is little knowledge about the entire process)<br>- Groundwater as a carbon sink<br>- Methane release from surface water bodies | Regulation of the concentrations of gases in the atmosphere and mediation of gaseous flows by aquatic ecosystems or the water itself, . . . | . . . that have an impact on global climate or oceans or offer protection to people |
| | | 2.2.6.2, 5.2.2.1 | Regulation of local temperature and humidity | - Lake breezes (movement of cool air from across the surface water body toward the land, altering the temperature in regions close to the water body)<br>- Frost occurrences in proximity to big lakes<br>- Springs | Mediation of ambient atmospheric conditions (including micro and mesoscale climates) such as local temperature and humidity, by virtue of the presence of aquatic organisms and abiotic water conditions, . . . | . . . that affect people's living conditions, well-being, or comfort |
| Support to other natural systems | | | | | Surface water bodies or aquifers forming input to the functioning of other natural systems . . . | . . . that provide regulating services |

**Table A3.** Cultural services. The water clause and use clause define the class.

| Division | Group | Code Used in CICES v 5.1 | Class | Examples of Services | Water Clauses | Use Clauses |
|---|---|---|---|---|---|---|
| Direct, in-situ and outdoor interactions that depend on presence in the environmental setting | Physical and experiential interactions with the natural environment | 3.1.1.1, 3.1.1.2, 6.1.1.1 | Activities promoting health, recuperation, or enjoyment through active, immersive, passive, or observational interactions | - Tourism and recreation through fishing, swimming, nature watching, cave tourism, skating, and skiing<br>- Tourism and recreation through canoeing, sailing, etc. | The abiotic or biophysical characteristics of water or the qualities of aquatic organisms or ecosystems . . . | . . . that enable active, or passive, physical and experiential interactions such as use, enjoyment, view, or observation |
| | Intellectual and representative interactions with natural environment | 3.1.2.1, 3.1.2.2, 6.1.2.1 | Scientific investigation, creation of traditional knowledge, education, training | - Scientific studies<br>- Outdoor education and excursions<br>- Springs used for environmental monitoring | The abiotic or biophysical characteristics of water or the qualities of aquatic organisms or ecosystems . . . | . . . that are the subject matter for in-situ research, teaching, or skill development |
| | | 3.1.2.3, 6.1.2.1 | Culture or heritage | - Historical understanding through, e.g., agricultural landscapes, historical activities, e.g., log driving and historical artifacts, e.g., water mills | The abiotic or biophysical characteristics of water or the qualities of aquatic organisms or ecosystems . . . | . . . that contribute to cultural heritage or historical knowledge |
| | | 3.1.2.4, 6.1.2.1 | Aesthetic experiences | - Sites of great beauty | The abiotic or biophysical characteristics of water or the qualities of aquatic organisms or ecosystems . . . | . . . that are appreciated for their inherent beauty |
| Indirect, remote, often indoor interactions that do not require presence in the environmental setting | Spiritual, symbolic, and other interactions with the natural environment | 3.2.1.1, 3.2.1.2, 6.2.1.1 | Religious, sacred, or symbolic meaning | - Sacred springs | The abiotic or biophysical characteristics of water or the qualities of aquatic organisms or ecosystems . . . | . . . that have symbolic or spiritual importance such as being recognized by people for their cultural, historical, or iconic character and that are used as emblems or signifiers of some kind, or being deemed to have sacred or religious significance for people |
| | | 3.2.1.3, 6.2.1.1 | Entertainment or representation | - Possibility for amusement or enjoyment via different media | The abiotic or biophysical characteristics of water or the qualities of aquatic organisms or ecosystems . . . | . . . that provide material or subject matter that can be communicated to others via different media for amusement or enjoyment |

**Table A3.** *Cont.*

| Division | Group | Code Used in CICES v 5.1 | Class | Examples of Services | Water Clauses | Use Clauses |
|---|---|---|---|---|---|---|
| | Other biotic or abiotic characteristics that have a non-use value | 3.2.2.1, 3.2.2.2, 6.2.2.1 | Existence, bequest, or option value | - Existence of species, landscape elements, etc. | The abiotic or biophysical characteristics of water or the qualities of aquatic organisms or ecosystems . . . | . . . that people seek to preserve because of their non-utilitarian qualities or importance to others and future generations |
| Support to other natura systems | | | | | Surface water bodies or aquifers forming input to the functioning of other natural systems . . . | . . . that provide cultural services |

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
