# Peer review of "Integrating Ecosystem Services into Risk Assessments for Drinking Water Protection"

_water, doi:10.3390/w14081180_

Round 1
Reviewer 1 Report
The main point of view in doing a water safety plan is drinking water safety for humans. This paper seems to present a novel research adding to water safety planning (WSP) a new aspect the ecosystem services which will most likely be the way ahead. It will though make the WSP even more complicated for the small or even medium size water supplies to carry out and this is where most of the non-compliance to drinking water regulation occur.
My few comments are as follows:
Chapter 2 explains ES and WSS well but would benefit from being shorter.
Text in figures and tables are often too small to read, e.g., in Figure 5 and legends in Table 6.
In line 141 water source services it written two times.
Reviewer 2 Report
The article deals with integrating ecosystem services into risk assessment specifically for drinking water protection, which is a new approach.
The methodology developed is well described and supported with supplement material so that every step is documented and easy to follow.
The application on the case study provides a better understanding of the methodology and proves that the methodology is applicable, with inside on the limitations and providing recommendations for further research and methodology improvement.
Suggested corrections to the article:
The text in Figure 5. is difficult to read, a more simplified scheme should be prepared.
In Figure 7. title, the max score of 125 should be explained since 0 is a score in the matrix for some hazard-service pairs, but there is no risk score of 125. The sentence could just be reformulated to explain the range of risk scores.
Reviewer 3 Report
This work proposes a list of 17 water system services (WSS) that allows assessment of all biotic and abiotic services provided by 18 drinking water source. Also, the manuscript illustrate how the ES framework can be integrated into risk assessments for drinking water protection to ensure that the full range of services provided by the water source is accounted for in decision making.
General comments
The manuscript contains descriptions of studies that have been well planned, carried out and interpreted. There are sufficient details given for integrating ecosystem services into a risk assessment for drinking water protection, including the development of the region-specific list of WSS, how it is applied to identify and quantify services, and how it is integrated into a risk assessment.
In my opinion, the conclusions could be improved in accordance with the case study made by the authors.
Reviewer 4 Report
Dear author,
your contribution is well written, the methodology is presented in a simple and comprehensive way !
I have only one suggestion : you should add in the lietrature review and in the discussion section a paragraph to explain how the proposed methodology deals with the concept of IWM ( integrated water amangement) and the short water cycle? These concepts merit to be cited
